# Research on Adolescents Regarding the Indirect Effect of Depression, Anxiety, and Stress between TikTok Use Disorder and Memory Loss

**DOI:** 10.3390/ijerph18168820

**Published:** 2021-08-21

**Authors:** Peng Sha, Xiaoyu Dong

**Affiliations:** School of Journalism and Communication, Southwest University, Chongqing 400715, China; dongxy@swu.edu.cn

**Keywords:** TikTok use disorder, depression, anxiety, stress, digit span

## Abstract

This research involved the participation of 3036 Chinese students in the first and second years of senior high school. The adolescents were active users of TikTok. The mediating effect of depression, anxiety, and stress between TikTok use disorder and memory loss was investigated. A forward and backward digit span test was applied to measure memory loss. Structural equation modeling (SEM) was established, and SPSS Amos was used for analysis. The results show a partial mediation effect of depression and anxiety between TikTok use disorder and forward digit span. A partial mediation effect of depression, anxiety, and stress between TikTok use disorder and backward digit span is also shown. These results also show gender differences. Attention should be given to male students, who have more depression, anxiety, and stress than female students; they also have more memory loss.

## 1. Introduction

### 1.1. Internet, Smartphone, or TikTok Use Disorder

The concept of “non-chemical addiction” was introduced in 1990 [1]. At that time, engaging in excessive online activities such as online sex and Internet games was initially called Internet addiction [2]. In 1990, only about 250 behavioral addiction papers were published, while in 2013, 2563 papers were published. General information, social networking, email, chat, videos, and films were reported to be the most popular online activities of Internet users [3]. Internet use disorder has a strongly negative influence on normal psychological development, and it can lead to syndromes such as stress [4]. Some researchers have investigated Internet communication in terms of the use of social networking sites such as WhatsApp and Facebook [5,6]. Easy access to the Internet with smartphones increased the popularity of social networks [7]. Very early research on the addictive use of the Internet was published in 1996 [8]. Many researchers prefer to use the term “Internet use disorder” [5,9] or “smartphone use disorder” [10,11,12], instead of “addiction”; however, these terms have still not been accepted by ICD-11 or DSM-5 [13].

TikTok is the most popular smartphone application of Chinese origin in the world. The total number of active TikTok users worldwide is 1.5 billion, and most of them are teenagers [14]. According to a report at the end of 2020, the number of monthly active TikTok users worldwide was 800 million. About 81% of Chinese users were young people under 35 years old [15]. The gratification of entertainment is the main driver for TikTok users. Adolescent groups are more active on the application because of their identity-creation and social contact needs [16]. Some researchers have indicated that self-expression and use satisfaction are associated with the motivations of TikTok users [17].

### 1.2. Internet or Smartphone Use Disorder and Depression, Anxiety, and Stress

According to a report by the WHO [18], the prevalence of depression and anxiety was 4.4% and 3.6%, respectively. Generally, females have a higher prevalence of depression and anxiety than males. In adolescence, the prevalence of depression and anxiety reaches the highest point [19]. Junior high school students, because of the high academic pressure and their monotonous life, are vulnerable to mental health problems such as depression [20].

Depression and social anxiety in junior high school students can be used as a mediator to explain the relationship between Internet use disorder and maladaptive cognition [21]. In research on adolescents [22], stress was highly associated with social anxiety. Social anxiety can act as a mediator between Internet use disorder and stress. Research on junior college students with an average age of 17 years [23] investigated the relationship between problematic Internet use, depression, anxiety, and stress. The more problematic the Internet use, the heavier the depression, anxiety, or stress. Moreover, depression, anxiety, and stress were positively associated with each other.

The relationship between anxiety, depression, and smartphone use disorder has been researched [24]. Depression and anxiety were shown to be highly correlated, and a positive correlation was found between anxiety and smartphone use disorder, but not between depression and smartphone use disorder. Some studies [25,26,27,28,29] showed that the use of social networking sites was positively linked to depression. However, other studies [30] showed that there was no correlation between the use of social networking sites and depression. Between users and non-users of Facebook, no differences were found with regard to depression, anxiety, and stress [31]. Internet use expectancies and dysfunctional cognition, such as suppression, maladaptive problem-solving, and avoidance, can be regarded as mediators between Internet use disorder and psychopathological aspects such as depression and social anxiety [32].

### 1.3. Depression, Anxiety, Stress, and Working Memory Capacity

The influence of depression on memory has been researched [33]. In this research, the influence of depression on memory varied by age and gender. The relationship between anxiety and working memory capacity, as an element of fluid cognition, has been researched [34]. The causal pathways from anxiety to low working memory were established. Furthermore, low working memory was found to have an effect on cognitive vulnerability, which has a feedback effect on anxiety. The relationship between anxiety, working memory, and gender has been researched [35]. State anxiety was found to vary by gender. However, a gender effect on trait anxiety was not found. Visual working memory was positively linked to math anxiety. However, there was no significant correlation between visual working memory and state anxiety or trait anxiety. A positive correlation between anxiety and stress was found [36]. Both anxiety and stress were negatively linked to visuospatial working memory, but they were not linked to verbal working memory, although there was a strong correlation between visuospatial and verbal working memory. The relationship between stress and working memory capacity has been researched [37]. Stress was found to be negatively linked to working memory. However, a correlation between state anxiety and working memory was not found. Some researchers [38] found a correlation between depression, anxiety, and working memory capacity, but found that situational stress had no influence on working memory capacity.

A correlation between forward and backward digit span was reported. Difficulties with forward and backward digit span in children were linked to learning disorders [39]. Brener [40] conducted an experimental investigation of memory span, including a list of materials with increasing difficulty. Digit span had a lower difficulty level than consonants, colors, and words. The influence of depression on reading span and word span has been researched [41]. A lower capacity of reading and word span by depressed patients was shown, and reading span was shorter than word span for both depressed and non-depressed persons. Digit span as a subtest of WAIS-IV was used to search its relationship with depression and anxiety. However, no significant correlation was found in this study [42]. Some researchers [43] did not find any correlations between forward or backward digit span and state anxiety, trait anxiety, or stress.

### 1.4. Internet or Smartphone Use Disorder and Working Memory Capacity

Correlations between use disorder, smartphone use disorder, and working memory have been researched [44]. Smartphone use disorder was found to be highly linked with Internet use disorder. Both Internet and smartphone use disorder were found to be negatively linked with working memory. The correlation between smartphone use disorder and working memory was found to be stronger than the correlation between Internet use disorder and working memory. However, this research did not find a gender effect on Internet or smartphone use disorder and working memory capacity. The mediating effect of depression, anxiety, and stress between problematic social media use and memory capacity was researched [45]. In this research, the Memory Awareness Rating Scale (MARS-MPS) was used to evaluate memory performance. The PROCESS macro in SPSS was used to analyze the pathways. This study was based on adults with an average age of about 30, and there were 466 valid participants. The results showed that only anxiety had a partial mediating effect between problematic social media use and memory performance.

### 1.5. Hypotheses

In this present study, the hypotheses are as follows (Figure 1):

**Hypothesis** **1** **(H1):**
*TikTok use disorder (TTUD) is positively linked to memory loss.*


**Hypothesis** **2** **(H2):**
*TTUD is positively linked to depression, anxiety, and stress.*


**Hypothesis** **3** **(H3):**
*Depression, anxiety, and stress are positively linked to memory loss.*


**Hypothesis** **4** **(H4):**
*Depression, anxiety, and stress have a mediating effect between TTUD and memory loss.*


## 2. Materials and Methods

### 2.1. Participants

The participants in the study were 3036 Chinese students in the first and second year of senior high school. Their participation was voluntary and anonymous (Table 1).

### 2.2. Measurement Instruments

The Smartphone Addiction Scale, Short Version (SAS-SV) [46] was used in this study. TTUD was adapted from the SAS-SV, in which “smartphone” was changed to “TikTok”. This questionnaire consists of 10 items, rated on a 6-point Likert-type scale, ranging from “strongly disagree” coded as 1, to “strongly agree” coded as 6. Higher scores indicate a higher risk of TikTok use disorder. In this study, the Cronbach’s alpha coefficient of TTUD was 0.91.

The Depression Anxiety Stress Scales 21 (DASS-21) [47,48] was used in this study. This questionnaire consists of 21 items rated on 4-point Likert scale, from 0 for “did not apply to me at all” to 3 for “applied to me very much”. Groups of 7 items are used to measure depression, anxiety, and stress. Higher scores indicate more severe symptoms. Cronbach’s alpha of depression was 0.88, anxiety was 0.86, and stress was 0.87.

Forward and backward digit spans were tested to measure memory loss. A number was given at random, beginning with a 2-digit number. The digits were increased until a wrong answer was given. A number was not repeated until a 10-digit number was reached; for example, 112 or 121, in which 1 was repeated, would not happen. The result was regarded as valid for the forward digit span test when the answer was a 3-digit to 11-digit number, and for the backward digit span test when the answer was a 2-digit to a 9-digit number.

### 2.3. Statistic Software

For this study, SPSS Amos 24.0 and SPSS 26.0 (IBM: New York, NY, USA) were used.

## 3. Results

Descriptive information divided by gender is shown in Table 2.

Pearson’s correlations between the research variables are shown in Table 3.

The results of pathway analysis are shown directly in Figure 2, Figure 3, Figure 4, Figure 5, Figure 6 and Figure 7. First, the results of forward and backward digit span tests of all participants, male and female, were calculated (Figure 2 and Figure 3). Second, the tests of male participants were calculated (Figure 4 and Figure 5). Finally, the tests of female participants were calculated (Figure 6 and Figure 7). Pathway coefficients shown in the figures are unstandardized.

For all participants, male and female, based on the forward digit span test, depression and anxiety have a partial mediating effect between TTUD and forward digit span memory capacity. Based on the backward digit span test, depression, anxiety, and stress have a partial mediating effect between TTUD and backward digit span memory capacity. For male participants, based on both types of digit span tests, only depression and anxiety showed partial mediating effects. For female participants, the partial mediating effects were the same as for all participants.

The criteria for good model fit were as follows: Chi-square/df < 5, RMSEA < 0.08, CFI > 0.95, and TLI > 0.95 [49]. Model fit information of the structural models in this study is shown in Table 4.

## 4. Discussion

This study was aimed at examining the mediating effect of depression, anxiety, and stress between TTUD and memory loss, focused on adolescents and divided by gender. TTUD was higher for female participants than male participants. This was the same as in early research on smartphone use disorder [12]. TTUD is positively linked with memory loss (H1). The greater the memory loss, the smaller the digit span. This result is gender independent. In this study, male participants showed more depression, anxiety, and stress than female participants. Kessler [50] indicated that throughout the lifespan, the prevalence of depression and anxiety in women is 1.5 times higher than in men. Taking a closer look at the gender effect on these symptoms, it did not differ between these results. This study was cross-sectional; prevalence throughout the lifespan was totally independent.

Some researchers have indicated that the gender effect could vary by factors such as age [51], anamnesis [52], or other mediating effects [53]. Gao [54] investigated college students and found no significant difference in depression or stress in first-year students and no significant difference in depression, anxiety, or stress in third-year students of different genders. TTUD is positively linked to depression, anxiety, and stress (H2). This result agrees with those of other researchers [23]. Depression, anxiety, and stress are positively linked to memory loss (H3). However, this hypothesis was not proven with regard to stress on the forward digit span test; it was proven with the backward digit span test, except for stress on male participants. Depression, anxiety, and stress have a mediating effect between TTUD and memory loss (H4). The partial mediating effect of depression and anxiety was proven with the forward digit span test, and stress had no mediating effect. However, with the backward digit span test, all three symptoms had a partial mediating effect, except for stress on male participants.

This research investigated a homogeneous group of participants from a normal senior high school in China. This sample was not representative of all adolescents. Generalizing the results will depend on further research. However, attention should be given to male students at senior high schools in China. Although their TTUD scores were lower than those of female students, they suffered more depression, anxiety and stress and had more memory loss than female students. For further studies, longitudinal research would be interesting. Because of the limitation of the cross-sectional design, a conclusion could not be made, as the more severe memory loss in male students resulted from the additional depression, anxiety, and stress they suffered. Furthermore, due to the cross-sectional study design, causal relationships between research variables could not be determined. To determine causal relationships, more information or more research would be needed. Some researchers [6] regarded depression, anxiety, and stress as independent variables and Internet use disorder as the dependent variable. The effects within or causal relationships between these variables may be reciprocal.

## 5. Conclusions

TikTok use disorder (TTUD) is positively linked to memory loss, and it is also positively linked to depression, anxiety, and stress. Depression, anxiety, and stress are positively linked to memory loss. Furthermore, depression, anxiety, and stress have a mediating effect between TTUD and memory loss.

A partial mediation effect of depression and anxiety between TTUD and forward digit span is shown. A partial mediation effect of depression, anxiety, and stress between TTUD and backward digit span is also shown. These results also show gender differences. Attention should be given to male students, who have more depression, anxiety, and stress than female students; they also have more memory loss.

## Figures and Tables

**Figure 1 ijerph-18-08820-f001:**
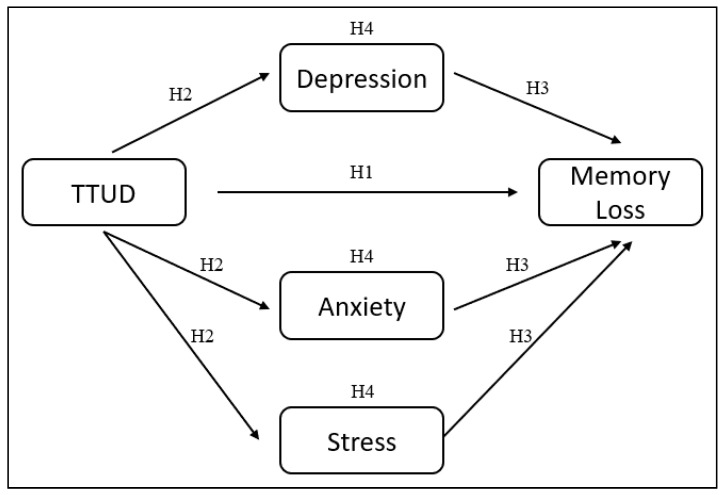
Structural model of hypotheses.

**Figure 2 ijerph-18-08820-f002:**
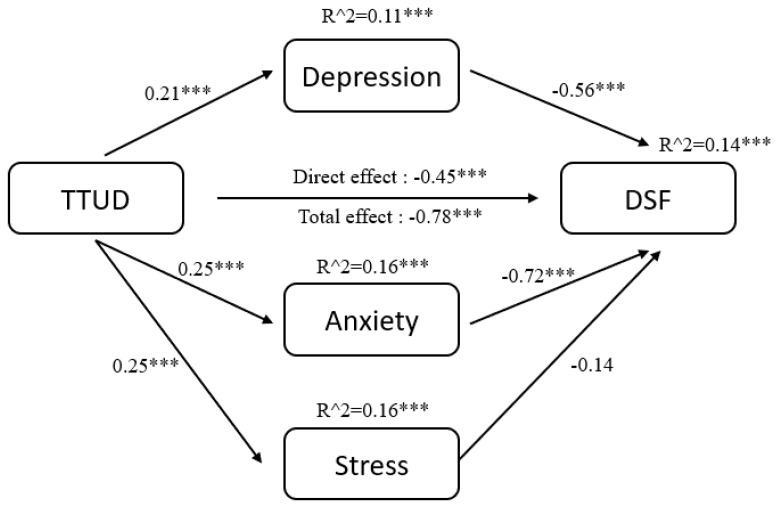
Structural model of forward digit span test for male and female participants. TTUD, TikTok use disorder; DSF, forward digit span. *** *p* < 0.001.

**Figure 3 ijerph-18-08820-f003:**
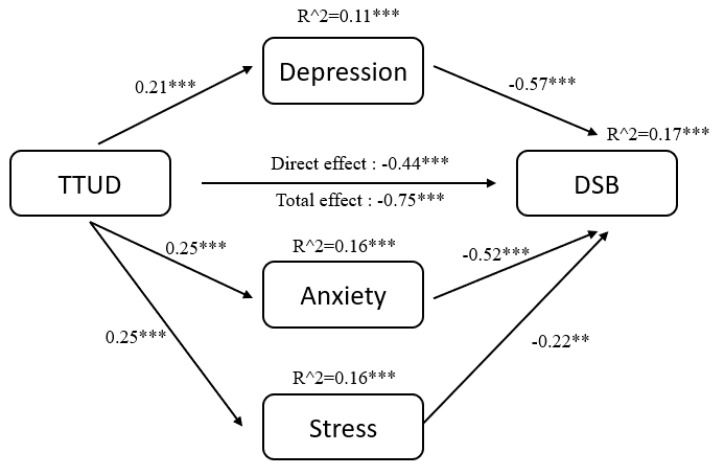
Structural model of backward digit span test for male and female participants. DSB, backward digit span. ** *p* < 0.01, *** *p* < 0.001.

**Figure 4 ijerph-18-08820-f004:**
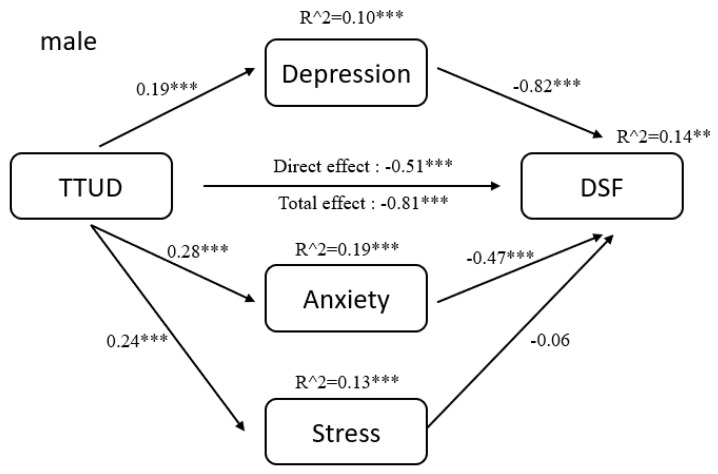
Structural model of forward digit span test for male participants. ** *p* < 0.01, *** *p* < 0.001.

**Figure 5 ijerph-18-08820-f005:**
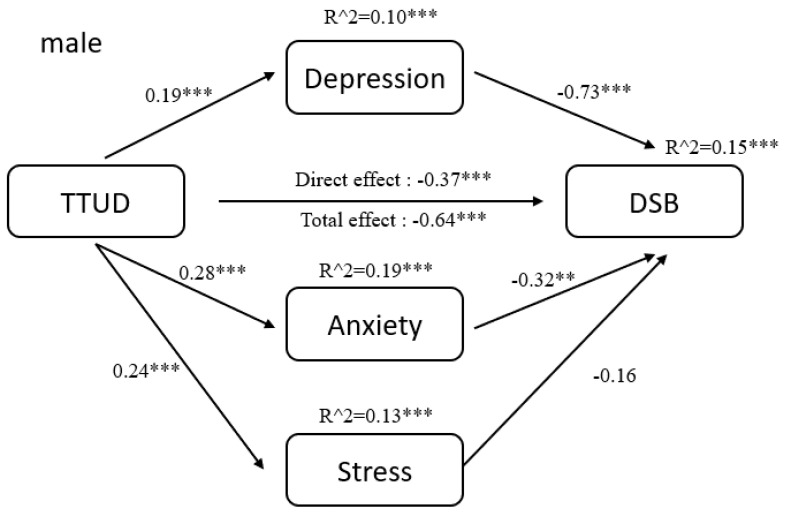
Structural model of backward digit span test for male participants. ** *p* < 0.01, *** *p* < 0.001.

**Figure 6 ijerph-18-08820-f006:**
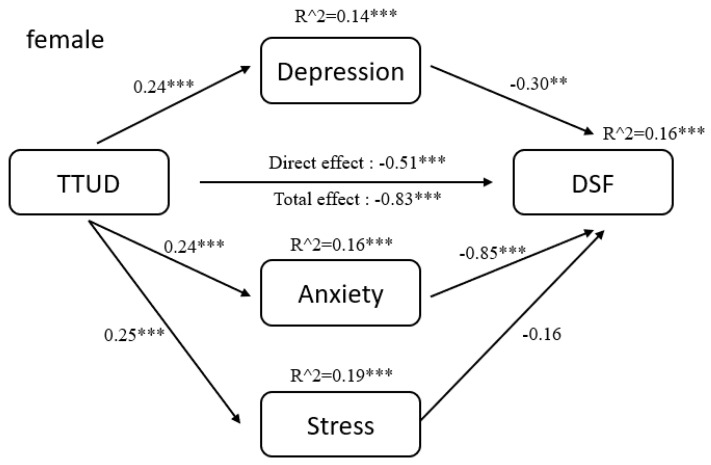
Structural model of forward digit span test for female participants. ** *p* < 0.01, *** *p* < 0.001.

**Figure 7 ijerph-18-08820-f007:**
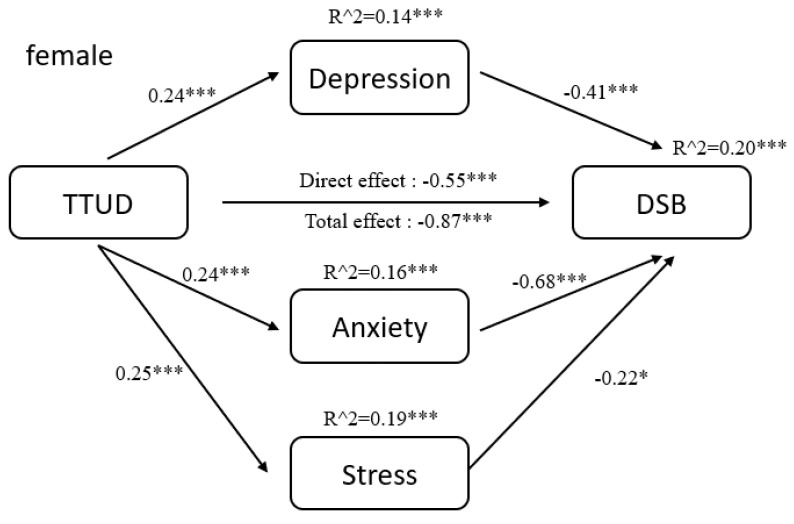
Structural model of backward digit span test for female participants. * *p* < 0.05, *** *p* < 0.001.

**Table 1 ijerph-18-08820-t001:** Participants’ information.

Gender	Sample Size	Proportion of Gender	Mean Age	SD of Age
Total	3036	100%	16.56	0.62
Male	1305	43%	16.61	0.60
Female	1731	57%	16.52	0.64

**Table 2 ijerph-18-08820-t002:** Descriptive statistics divided by gender.

	Male		Female			
	Mean	SD	Mean	SD	*T*	*p*
Age	16.61	0.60	16.52	0.64	3.86	<0.001
TTUD	34.92	9.98	37.10	9.62	−6.08	<0.001
Depression	5.65	4.05	5.09	3.95	3.77	<0.001
Anxiety	6.68	4.11	5.55	3.75	3.60	<0.001
Stress	6.98	4.41	6.66	4.17	2.06	0.04
DSF	7.25	2.10	7.79	2.04	−7.19	<0.001
DSB	5.20	1.70	5.49	1.85	−4.50	<0.001

TTUD, TikTok use disorder; DSF, forward digit span; DSB, backward digit span.

**Table 3 ijerph-18-08820-t003:** Pearson’s correlations between research variables.

	Mean	SD	2	3	4	5	6
1 TTUD	36.17	9.83	0.27 **	0.33 **	0.33 **	−0.28 **	−0.31 **
2 Depression	5.33	4.01		0.51 **	0.49 **	−0.29 **	−0.32 **
3 Anxiety	5.78	3.92			0.50 **	−0.31 **	−0.32 **
4 Stress	6.80	4.28				−0.25 **	−0.28 **
5 DSF	7.56	2.08					0.33 **
6 DSB	5.37	1.79					

** *p* < 0.01.

**Table 4 ijerph-18-08820-t004:** Model fit information.

	Chi-Square/df	CFI	TLI	RMSEA
Male and female DSF	4.573	0.962	0.958	0.034
Male and female DSB	4.564	0.962	0.959	0.034
Male DSF	2.498	0.962	0.965	0.034
Male DSB	2.504	0.965	0.962	0.034
Female DSF	2.822	0.964	0.961	0.032
Female DSB	2.847	0.964	0.961	0.033

## Data Availability

The datasets generated during and analyzed during the current study are available from the corresponding author on reasonable request.

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
