# Peer review of "Research on Adolescents Regarding the Indirect Effect of Depression, Anxiety, and Stress between TikTok Use Disorder and Memory Loss"

_ijerph, 2021, doi:10.3390/ijerph18168820_

Round 1

Reviewer 1 Report

Review of the Research Article:" Research of adolescents on the indirect effect of depression, anxiety and stress between TikTok use disorder and memory loss"

Solving the problems of computer addiction, which belongs to the category of "non-chemical addictions", is one of the global medical and social problems for the whole world.

Important tasks today are the issues of prevention and treatment of behavioral and mental disorders associated with excessive use of Internet resources. The research conducted by the author concerns the solution of these rather complex and problematic aspects, which determines the relevance of the conducted research.

The authors of the work focuses on the preventive direction, which can be fundamental in the development of a set of medical measures. The effectiveness of prevention with the inclusion of substitution therapy has been proven, which allows optimizing the provision of medical care to this specialized contingent.

The work was performed at a sufficiently high methodological level, with a sufficient sample volume, correct application of statistical processing methods and is a complete scientific study dedicated to solving one of the important tasks of medical science in the field of addiction problems.

The article title and abstract are appropriate. The purpose of the article and its significance is stated clearly. The study methods are sound and appropriate. The writing is clear and concise. The summary is accurate and supported by the content. The article is of interest to members of the education research community.

I believe that the Research Article:"Research of adolescents on the indirect effect of depression, anxiety and stress between TikTok use disorder and memory loss" can be recommended for publishing.

Author Response

Thank you very much from all my heart for your review and opinions. I appreciate your comments very much. Thank you very much for the high quality of your academic standard and understanding in the research work. Your opinions have so much encouraged my work!

Moderate English changes has been done.

Reviewer 2 Report

This is a very hard paper to follow. The introduction should build to the stated hypotheses, so it is unclear how the hypotheses develop from what is presented in the Introduction. There is no description on the ethical consent of subjects, inclusion, or exclusion criteria. Without inclusion/exclusion criteria all variables are not controlled. Table 1 and 2 are redundant with regard to age. Sex is not distinguishing factor. There is nothing that is significantly different in males, but not in the females, in some cases the values are the same.  Was the DASS-21 modified for use in adolescents?  

Round 2

Reviewer 2 Report

The authors have addressed my previous concerns.